# miR-146a-5p Promotes Angiogenesis and Confers Trastuzumab Resistance in HER2+ Breast Cancer

**DOI:** 10.3390/cancers15072138

**Published:** 2023-04-04

**Authors:** Paula Cabello, Sandra Torres-Ruiz, Anna Adam-Artigues, Jaume Forés-Martos, María Teresa Martínez, Cristina Hernando, Sandra Zazo, Juan Madoz-Gúrpide, Ana Rovira, Octavio Burgués, Federico Rojo, Joan Albanell, Ana Lluch, Begoña Bermejo, Juan Miguel Cejalvo, Pilar Eroles

**Affiliations:** 1Biomedical Research Institute INCLIVA, 46010 Valencia, Spain; 2International University of Valencia—VIU, 46002 Valencia, Spain; 3Department of Medical Oncology, University Clinical Hospital of Valencia, 46010 Valencia, Spain; 4Department of Pathology, Jiménez Díaz Foundation, 28040 Madrid, Spain; 5Center for Biomedical Network Research on Cancer (CIBERONC), 28040 Madrid, Spain; 6Department of Medical Oncology, Hospital del Mar, 08003 Barcelona, Spain; 7Cancer Research Program, IMIM (Hospital del Mar Medical Research Institute), 08003 Barcelona, Spain; 8Department of Pathology, University Clinical Hospital of Valencia, 46010 Valencia, Spain; 9Department of Medicine, University of Valencia, 46010 Valencia, Spain; 10Department of Physiology, University of Valencia, 46010 Valencia, Spain; 11Department of Biotechnology, Polytechnic University of Valencia, 46022 Valencia, Spain

**Keywords:** HER2+ breast cancer, resistance, miR-146a-5p, trastuzumab, exosomes

## Abstract

**Simple Summary:**

Resistance to anti-HER2+ therapy remains the main clinical challenge in the management of the HER2+ breast cancer subtype. The objective of our study was to analyze the involvement of microRNAs in the resistance to trastuzumab. We identified miR-146a-5p as the major dysregulated microRNA among parental and trastuzumab-resistant HER2+ breast cancer cells. The gain- and loss-of-function of this miRNA modulates resistance to trastuzumab in vitro, and elevated levels of miR-146a-5p in the primary tumor have been associated with a poor prognosis. In addition, exosomes from trastuzumab-resistant cells contain high levels of miR-146a-5p, and may reduce the effect of trastuzumab on sensitive cancer cells, increasing the expression of epithelial-to-mesenchymal transition markers and the capacities for migration and angiogenesis. The results of this study demonstrate for the first time the involvement of miR-146a-5p in resistance to trastuzumab, and suggest that exosomes play an important role in this process.

**Abstract:**

Trastuzumab treatment has significantly improved the prognosis of HER2-positive breast cancer patients. Despite this, resistance to therapy still remains the main clinical challenge. In order to evaluate the implication of microRNAs in the trastuzumab response, we performed a microRNA array in parental and acquired trastuzumab-resistant HER2-positive breast cancer cell lines. Our results identified miR-146a-5p as the main dysregulated microRNA. Interestingly, high miR-146a-5p expression in primary tumor tissue significantly correlated with shorter disease-free survival in HER2-positive breast cancer patients. The gain- and loss-of-function of miR-146a-5p modulated the response to trastuzumab. Furthermore, the overexpression of miR-146a-5p increased migration and angiogenesis, and promoted cell cycle progression by reducing CDKN1A expression. Exosomes from trastuzumab-resistant cells showed a high level of miR-146a-5p expression compared with the parental cells. In addition, the co-culture with resistant cells’ exosomes was able to decrease in sensitivity and increase the migration capacities in trastuzumab-sensitive cells, as well as angiogenesis in HUVEC-2 cells. Collectively, these data support the role of miR-146a-5p in resistance to trastuzumab, and demonstrate that it can be transferred by exosomes conferring resistance properties to other cells.

## 1. Introduction

Breast cancer (BC) is the most prevalent cancer in the world, and together with lung cancer, is the main reason for why women die of cancer [1]. Around 15–20% of breast tumors overexpress the HER2 receptor, and this subtype presents a poor prognosis and a low overall survival (OS) due to its high proliferation and ability to metastasize [2,3]. The main targeted treatment for HER2-positive (HER2+) BC patients is trastuzumab, a monoclonal antibody which directly neutralizes the amplified HER2 receptor and blocks its signaling [3,4]. This therapy has improved the prognosis of HER2+ BC in the last few decades. However, there is still a group of patients that experience resistance to this treatment [5,6,7]. Several genetic mutations are well-known to be related to trastuzumab resistance [8,9,10,11,12,13,14]. However, the implication of epigenetics in trastuzumab resistance still needs a deeper exploration.

MicroRNAs (miRNAs) are small, non-coding RNAs capable of regulating gene expression, and therefore, numerous cellular processes [15,16]. In a number of cancers, miRNAs have already shown promise, both as therapeutic agents and as biomarkers for prognosis and prediction [17,18,19,20,21,22,23]. Similar to this, basal-like and luminal BC subtypes have been shown to present molecular signatures of differentially expressed miRNAs [24,25] that can be used to categorize the statuses of the estrogen receptor (ER), progesterone receptor (PR), and HER2/neu receptor [26,27,28].

Moreover, exosomes are small extracellular vesicles of size 40–130 nm released by almost all cell types, that mediate intercellular communication. These vesicles carry different molecules from the releasing cell, mostly proteins, miRNAs, lipids, and metabolites for cell-to-cell communication. This process is dysregulated in many tumoral processes and can promote malignancy. It has been described that exosomes from tumor cells can activate angiogenesis [29,30] and alter the tumor microenvironment to increase its aggressiveness [31], or to form a metastatic niche in different types of cancer [29]. Exosomes can also mediate the horizontal transmission of resistance to chemotherapy between cells [32,33], suggesting a new and faster mechanism for transmitting resistance, other than cell division. Furthermore, exosomes can provide information about the tumor via a minimally invasive liquid biopsy procedure, and they might also be potential targets to inhibit their supporting function in the development of tumors [34,35].

This work aimed to study the miRNAs involved in trastuzumab resistance, as well as to its transmission by exosomes in HER2+ BC. To address this issue, we studied the differential expression between parental and acquired trastuzumab-resistant cell lines, and the most promising miRNA was also analyzed in HER2+ BC patients’ samples. We also evaluated its role in different pathways in order to describe its mechanism of action, as well as its presence in exosomes. The results show a new epigenetic mechanism leading resistance to trastuzumab through miR-146a-5p and its transmission by exosomes.

## 2. Materials and Methods

### 2.1. Cell Lines and Culture

The human HER2+ BC cell lines SKBR3, BT474, and the acquired trastuzumab-resistant SKBR3 (SKBR3r) and BT474 (BT474r) were obtained from FR’s group at the Foundación Jiménez Díaz Hospital (Madrid). Cell lines were grown in Dulbecco’s Modified Eagle medium nutrient mixture F-12 (DMEM/F12) with 2.5 mM L-Glutamine and 15 mM HEPES (Gibco, Thermo Fisher Scientific, Inc., Waltham, MA, USA) supplemented with 10% fetal bovine serum (FBS) and 1% antibiotics (100 U/mL penicillin and 100 mg/L streptomycin). SKBR3r and BT474r cells were generated via culture with 15 μg/mL trastuzumab for 6–12 months [36], and cultured with that constant concentration of trastuzumab. Cells were maintained in a humidified atmosphere at 37 °C with 5% CO_2_. Thermo Fisher Scientific provided BD Human Umbilical Vein Endothelial Cells (HUVEC-2) (#10683493, Corning, Corning, NY, USA), which were grown in EGM-2 medium (Lonza, Durham, NC, USA).

### 2.2. miRNA Microarray and miRNA Target Prediction

Agilent RNA 6000 Nano Assay LabChip^®^ and Agilent 2100 Bioanalyzer equipment were used to assess the integrity of total RNA (Agilent Technologies Inc., Santa Clara, CA, USA). Then, the microarray assay was carried out using an Affymetrix GeneChip™ miRNA 4.0 Array (Affymetrix, Santa Clara, CA, USA) following the manufacturer’s instructions. Affymetrix GeneChip™ miRNA 4.0 microarrays were prepared by the University of Valencia’s genomic and epigenetic section. Statistical analysis for evaluating differential expression was performed in R software using the samr package and applying the SAM (Significance Analysis of Microarrays) method. Significance cut-off was determined using the delta adjustment parameter, selected by the user based on the false positive rate (FDR). To analyze the specific targets of miRNAs, miRWalk2.0 software was used. These target genes were implemented in DIANA-miRPath 2.1 for showing the KEGG pathways most closely associated with them.

### 2.3. miRNA Mimics/Inhibitors and siRNA Transfection

A 50 nM concentration of miRNA mimic or inhibitor (hsa-miR-146a-5p mirVana^®^ mimic #MC10722 and hsa-miR-146a-5p mirVana^®^ inhibitor #MH10722, Ambion Inc., Austin, TX, USA) was transfected using Lipofectamine 2000 (Thermo Fisher Scientific) following the manufacturer’s instructions. A 50 nM concentration of scramble, a random sequence miRNA molecule, served as a miRNA negative transfection control (#4464059, Ambion). For mRNA knockdown, 100 nM of CDKN1A Silencer^®^ siRNA (#1624, Ambion) and CDKN1A#2 Silencer^®^ siRNA (#1531, Ambion) was transfected as explained. The transfection medium was replaced with complete medium after 6 h. All the experiments were carried out 48 or 72 h after transfection, except for the proliferation assay carried out at 0, 24, 48 and 72 h, or at 7 days for trastuzumab treatment.

### 2.4. Cell Viability Assay

SKBR3/SKBR3r and BT474/BT474r cells were seeded at an initial density of 3 × 10^3^ cells in 96-well plates and treated with 15 μg/mL trastuzumab (F. Hoffmann-La Roche Ltd., Basel, Switzerland) for 7 days with or without the prior transfection of miRNA mimics/inhibitors, silencers, or co-culture with exosomes. Cell viability was measured using an MTT-based Cell Growth Determination Kit (#GDC1, Sigma-Aldrich, Merck, Rahway, NJ, USA), following the manufacturer’s instructions. Absorbance was measured at 570 nm, with a background correction of 690 nm. The percentage of the number of cells from each group was referenced to the controls.

### 2.5. Clinical Samples

Formalin-fixed and paraffin-embedded (FFPE) samples from human HER2+ BC primary tumor tissues were selected through the Pathology Department of the Hospital Clínico de Valencia (n = 33) from the INCLIVA Biobank collection. Patients were treated with chemotherapy + anti-HER2 (trastuzumab ± pertuzumab) for 6 months prior to surgery. The clinical information is summarized in Table 1. The study was approved by the INCLIVA institutional review board (2018/077) and followed the ethical guidelines (Declaration of Helsinki). All participants provided written informed consent to take part in the study.

### 2.6. RNA Isolation

Total RNA from cell lines was extracted using TRIzol reagent (Invitrogen, Carlsbad, CA, USA) according to the manufacturer’s instructions. The total RNA from tissue block samples of BC patients was isolated using the RecoverAll Total Nucleic Acid Kit (Ambion). A NanoDrop spectrophotometer was used to check the purity ratios and to measure the RNA concentrations.

### 2.7. miRNA and Messenger RNA (mRNA) Expression via Real-Time Quantitative PCR (RT-qPCR)

A total of 1000 ng of RNA was retrotranscribed into cDNA using the High-Capacity cDNA Reverse Transcription Kit for genes (ThermoFisher Scientific) and the TaqMan MicroRNA Reverse Transcription Kit for miRNAs (ThermoFisher Scientific). miRNA and mRNA expression levels were analyzed using specific commercial Taqman probes from Applied Biosystems (ThermoFisher Scientific) (#000468 for miR-146a-5p, #Hs00355782 for CDKN1A, #Hs01030099 for CCNB1, #Hs00958111 for VIM, and #Hs01549976 for FN1), and the commercial solution TaqMan Gene Expression Master Mix (Applied Biosystems, Waltham, MA, USA) and TaqMan Universal MasterMIx II, no UNG (Applied Biosystems) for mRNA and miRNA, respectively. All samples were analyzed in triplicate, and the QuantStudio Thermocycler (Applied Byosistems) was programmed specifically. The Ct of every gene or miRNA of interest were normalized to its endogenous control (ΔCt). The endogenous control for genes was GAPDH (#Hs03929097, Applied Biosystems). Nucleolar RNA RNU43 and miR-16a (#001095 and #000391, respectively, Applied Biosystems) were used as the endogenous controls of miRNAs, and only miR-16a for miRNAs present in exosomes. Relative expression was quantified using the 2^−ΔΔCt^ method.

### 2.8. Exosome Isolation and Transmission Electron Microscopy (TEM) Analysis

Conditioned media was collected after 24 h, and cell debris was removed by centrifuging it at 2000× *g* for 30 min. The supernatant was transferred to a new tube (#355642, Beckman Coulter, Brea, CA, USA) and centrifuged at 10,000× *g* for 30 min at 4 °C using break at deceleration. The supernatant containing exosomes was centrifuged at 100,000× *g* for 90 min at 4 °C using a Beckman 70 Ti Rotor into the Beckman Optima IE-80K Ultracentrifuge, and the pellet was resuspended in 1 mL of 0.2 µm filtered phosphate buffered saline (PBS). Exosomes were subsequently washed at 100,000× *g* for 90 min, and after discarding the supernatant, the final pellet was resuspended in 100 µL of PBS and stored at −80 °C. Then, 10 µL volumes of exosome sample were immediately incubated in 4% paraformaldehyde for fixation for TEM preparation, which consisted of double staining the sample with 20 min of 0.5% uranyl acetate and 10 min of 3% lead citrate on a 400-mesh copper grid. A JEOL JEM1010 TEM was used at 80 kV, and an automatic exposure time was used to observe the grids containing the exosomes.

All experiments with exosomes were performed, adding exosomes at a concentration of 100 µg/mL to the cell cultures, and using exosome-depleted FBS (FBS was also ultracentrifuged at 100,000× *g* for 18 h for this purpose).

### 2.9. Exosome Internalization

Exosomes were labeled with DiD fluorescent lipophilic membrane dye (#D307, ThermoFisher Scientific) according to the manufacturer’s instructions. The labeled exosomes were resuspended in DMEM and added to the cell cultures at a final concentration of 100 µg/mL for 3 h. The cell cultures were previously exposed or not to 15 μg/mL of trastuzumab. The fluorescence of the cells was subsequently quantified via flow cytometry using a Flow Cytometer BD LSRFortessa™ (BD Biosciences, Franklin Lakes, NJ, USA). Data were analyzed with BD FACSDiva X-20 software (BD Biosciences).

### 2.10. Western Blot

Cells were lysed using RIPA Lysis and Extraction Buffer (ThermoFisher Scientific), including anti-protease and anti-phosphatase reagent (ThermoFisher Scientific), and sonicated in 5 pulses of 10 seconds at 40% amplitude for protein extraction. Protein quantification was performed using the commercial kit Pierce BCA Protein Assay Kit (ThermoFisher Scientific) following the manufacturer’s instructions. The proteins were resolved on a 10% SDS–polyacrylamide gel using 30 µg per well, and transferred onto a nitrocellulose membrane, then blocked for 1 h in 5% powder milk with 1× TBS and 0.1% Tween-20 (#P9416-100ML, Sigma-Aldrich), and incubated with the antibodies CDKN1A 1:1000 (#1672947S, Cell Signaling, Danvers, MA, USA), CCNB1 1:1000 (#4138S, Abcam, Cambridge, UK), CDH1 1:1000 (#610181, BD Biosciences), CDH2 1:1000 (#4061S, Cell Signaling), Fibronectin 1:1000 (#32419, Abcam), VIM 1:1000 (#550513, BD), VEGFR2 1:1000 (#ab39638, Abcam), Calnexins 1:1000 (#ab22595, Abcam), CD9 1:500 (#ab92726, Abcam), Tsg-101 1:1000 (#ab125011, Abcam), EpCAM 1:1000 (#324207, Biolegend, San Diego, CA, USA), and GAPDH 1:1000 (#MA5-15738, Invitrogen) diluted in 5% powder milk in 1× TBS-Tween (TBS-T) overnight at 4 °C with agitation. The membranes were washed in TBS-T 3 times for 5 min, and then incubated in the corresponding secondary antibody (anti-Rabbit (#1677074S, Cell Signaling) or anti-Mouse (#1677076S, Cell Signaling), diluted 1:2000 in 1% milk in TBST-T). Pierce TM ECL Western Blotting Substrate (ThermoFisher Scientific) was used for protein detection.

### 2.11. Cell Cycle Analysis

To analyze the cell cycle, 3 × 10^4^ cells were seeded in 6-well plates in triplicate for each condition. At 48 h after transfection, cells were harvested with trypsin, washed twice with PBS, and later fixed with 70% cold ethanol and incubated at −20 °C overnight. The cells were resuspended in 10 μg/mL propidium iodide and incubated for 30 min at room temperature before analysis via flow cytometry using a Flow Cytometer BD LSRFortessa™ (BD Biosciences). Data were analyzed using FlowJo V10 (FlowJo LLC, BD Biosciences).

### 2.12. Wound-Healing Assay

SKBR3 or SKBR3r were seeded at 100% confluence. After 16 h of serum starvation, a wound was performed with a pipette tip, and cells were washed and subsequently treated with either 50 nM miR-146a-5p mimic, 100 nM CDKN1A silencer, 100 µg/mL exosomes, or PBS in FBS-free culture medium. Cells were allowed to migrate, and pictures were taken in triplicate at 0 h and 48 h with a microscope at 5× magnification. Images were analyzed using Image J to measure the areas devoid of cells.

### 2.13. Angiogenesis Assay

A 120 µL volume of Matrigel (Corning, Corning, NY, USA) per well was plated in a 48-well plate. Each well was filled with 1 mL endothelial media or CM from SKBR3 miR-146a-5p mimic, siCDKN1, or scramble transfected cells, and 3 × 10^4^ HUVEC-2 cells were seeded on top in a spiral motion for even distribution. Each condition was assayed in triplicate, and 5 pictures per well were taken after 8 h at 10× magnification. ImageJ was used for the analysis. Using the straight-line tab, the networks were traced out, and the nodes and lengths were counted and measured, subsequently adding the results from each image. The average for each well was calculated.

### 2.14. Survival Analysis

The prognostic value of miR-146a-5p was evaluated in a cohort of patients from the Hospital Clínico de Valencia, and in 608 HER2+ BC patients from the METABRIC database, using the Kaplan-Meier Plotter (https://kmplot.com/analysis/ (accessed on 25 March 2020)) [37]. Patients were classified based on the median miR-146a-5p expression in primary tumor samples, and Kaplan-Meier curves were plotted. The *p*-value was calculated using the Log-rank (Mantel-Cox) test.

### 2.15. TCGA Data Analysis

The data of *CDKN1A* and *CCNB1* expression were obtained from the University of California Santa Cruz (UCSC) Xena Cancer Browser database (https://xenabrowser.net (accessed on 25 January 2021)). In total, 333 TCGA BC patients were classified based on the median miR-146a-5p expression in tumor samples, and *CKDN1A* and *CCNB1* expression were represented. The *p*-value was calculated using the Mann-Whitney test, and normality was verified using the Shapiro-Wilk test.

### 2.16. Statistical Analysis

An analysis of the results was carried out using R and GraphPad Prism 8. All data were presented as mean ± standard deviation (SD). The Shapiro-Wilk test was performed to test for normality. Mean comparisons were performed using a two-tailed Student’s *t*-test for normal distribution, and the Mann-Whitney U test for abnormal distribution. An ANOVA test was used to identify the miRNAs differentially expressed between SKBR3r and SKBR3, or BT474r and BT474, from the Affymetrix GeneChip™ miRNA 4.0 Array data. Every experiment was performed in technical and biological triplicate for each condition. *p*-values of under 0.05 were considered to be statistically significant.

## 3. Results

### 3.1. miR-146a Is Upregulated in Trastuzumab-Resistant Cells

A total of 6631 human miRNAs/pre-miRNAs were analyzed for expression using the Affymetrix GeneChip™ miRNA 4.0 Array for SKBR3/SKBR3r and BT474/BT474r cells. RNA integrity and quality were evaluated, and every sample used showed a RIN number of higher than 8. The Kernel density estimator showed a high symmetry for our cell samples (Appendix A). Furthermore, box plots of the expression variability inside each array and across the samples showed excellent symmetry, with their median values being around 1. (Appendix A). Biological triplicates for each BC cell line presented a similar miRNA profile and clustered together in the heatmap representation. The ANOVA test evidenced significant differences in 265 miRNAs expressed between SKBR3 and SKBR3r, while 261 were differentially expressed in the case of BT474 and BT474r (*p* < 0.05). After adjusting for biologically significant fold-change (linear fold-change resistant versus parental > 2 and <−2) 4 and 16 miRNAs remained biologically significant in SKBR3/SKBR3r and BT474/BT474r, respectively (Figure 1A). From these significantly dysregulated miRNAs, MIMAT0000449 (miR-146a-5p) had the biggest fold-change (5.1 times higher) in SKBR3r compared to SKBR3, and was a commonly dysregulated miRNA in SKBR3 and BT474 cells (Appendix A). miR-146a-5p differential expression between the parental and resistant cell lines obtained from the array (Figure 1B) was confirmed via RT-qPCR (*p* = 0.0002 for SKBR3 and *p* < 0.0001 for BT474) (Figure 1C).

### 3.2. High Expression of miR-146a-5p Associated with Relapse and Shorter Disease-Free Survival in HER2+ BC Patients

To test the hypothesis of the potential association of miR-146a-5p overexpression with resistance to trastuzumab, its expression was analyzed in a cohort of HER2+ BC patients’ samples from the Hospital Clínico de Valencia. Patients who relapsed after the trastuzumab-containing treatment showed significantly increased miR-146a-5p expression compared to patients who remained disease-free after therapy (*p* = 0.040) (Figure 2A). Kaplan-Meier curves were also assessed to analyze their prognostic value, and the results showed a significantly shorter disease-free survival (DFS) in HER2+ BC patients with a high expression of miR-146a-5p, compared to those with lower miR-146a-5p levels (Hazard ratio (HR) = 2.677, 95% Confidence Interval (CI) 1.040–6.018, *p* = 0.043) (Figure 2B). Patients with a high expression of miR-146a-5p also showed a lower OS than patients with a low expression, although it did not reach significance (HR = 3.494, 95% CI 0.865–9.368, *p* = 0.086) (Appendix A). The OS analysis of the miRNA expression data available from the METABRIC database showed the same trend observed in our patients’ cohort. (HR = 1.31, 95% CI 0.99–1.73, *p* = 0.056) (Appendix A).

### 3.3. miR-146a-5p Expression Modulates the Trastuzumab Response in HER2+ BC Cell Lines

In order to evaluate trastuzumab response, we first performed proliferation curves with increasing doses of the drug in trastuzumab-resistant and parental cells. The calculated IC_50_ was significantly higher for SKBR3r and BT474r cells compared to their parental counterparts (*p* = 0.025 for SKBR3r and *p* = 0.014 for BT474r), thus confirming the resistance characteristics of these cells (Appendix A).

To analyze the potential involvement of miR-146a-5p in the trastuzumab response, parental and resistant cell lines were transfected with either a miR-146a-5p mimic or inhibitor, respectively, and treated or not with 15 μg/mL trastuzumab for 7 days. The results showed that SKBR3 and BT474 cells with miR-146a-5p overexpression significantly reduced its response to trastuzumab (*p* = 0.011 and *p* = 0.009, respectively) (Figure 3A). Oppositely, SKBR3r and BT474r cell lines transfected with miR-146a-5p inhibitor significantly increased its response to trastuzumab (*p* = 0.027 and *p* = 0.018, respectively) (Figure 3B). The gain- and loss-of-function of miR-146a-5p was confirmed in each experiment (Appendix A) at 48 h and after 7 days (Appendix A).

### 3.4. miR-146a-5p Overexpression Increases Migration, Angiogenesis, and Proliferation

Given that the resistance to trastuzumab has been previously associated with a mesenchymal-like phenotype [38,39], we next studied the effect of the miR-146a-5p gain-of-function on the epithelial-to-mesenchymal transition (EMT). The results showed that the transfection of the miR-146a-5p mimic was able to significantly increase the migration capacity in SKBR3 cells (*p* = 0.032) (Figure 3C,D). Moreover, it induced the expression of mesenchymal markers (N-cadherin, fibronectin, and vimentin) and reduced the epithelial markers (E-cadherin and EpCAM) (Figure 3E and Appendix A). In addition, the switch to a vascular phenotype has been related to the trastuzumab resistance of HER2+ BC [40] and the combined targeting of HER2 and VEGFR2, as a potential therapy for HER2+ BC [41]. Based on that, we performed an angiogenesis assay on HUVEC-2 cells treated with conditioned media (CM) from SKBR3 previously transfected with miR-146a-5p mimic or scramble. CM from the SKBR3 cells overexpressing miR-146a-5p significantly increased in angiogenesis (formation number of branches: *p* = 0.011; tubular length: *p* = 0.04) (Figure 3F,G), and induced the glycosylation of VEGFR2 (Figure 3H and Appendix A). Lastly, we also confirmed that miR-146a-5p gain-of-function significantly increased the cell proliferation rates of the SKBR3 and BT474 cell lines (Figure 3I), and a higher proliferation rate of SKBR3 was observed compared to the BT474 cell line (Appendix A). Contrarily, the results showed that miR-146a-5p loss-of-function decreased the cell proliferation rate of BT474r (Appendix A).

### 3.5. miR-146a-5p Overexpression Increases the Cell Cycle S and G2/M Phases

After analyzing the effect of miR-146a-5p on trastuzumab response and proliferation, we studied its interaction with candidate target genes in silico, in order to clarify their possible underlying mechanisms of action. Genes described as potential or already validated targets of miR-146a-5p were obtained from the miRWalk2.0 and Transcriptome Analysis Console (Affymetrix) software (Appendix A), and cell cycle was identified as one of the main interesting pathways among the miR-146a-5p described targets.

In order to confirm the effect of miR-146a-5p in the cell cycle, SKBR3 and BT474 cells were transfected with the miR-146a-5p mimic. miR-146a-5p overexpression significantly decreased the proportion of cells in the G0/G1 phase (*p* = 0.007 for SKBR3 and *p* = 0.037 for BT474), and increased the percentage of cells in the S (*p* = 0.042 for SKBR3 and *p* = 0.019 for BT474) and G2/M phases (*p* = 0.017 for SKBR3) (Figure 3J).

Next, we assessed the expression of *CDKN1A* and *CCNB1* genes, both described as putative miR-146a-5p targets, and involved in cell cycle regulation. An analysis of the BC samples from the TCGA dataset, ranked using the median miR-146a-5p expression, showed an inverse correlation of *CKDN1A* and a positive correlation of *CCNB1* expression with miRNA levels (Figure 4A). The BT474r and SKBR3r cell lines also showed lower levels of *CDKN1A* expression and higher levels of *CCNB1* compared to BT474 and SKBR3 (Figure 4B).

The negative regulation of CDKN1A and the positive regulation of CCNB1 by miR-146a-5p were confirmed at both the gene and protein levels. In addition, CDKN1A downregulation gave rise to CCNB1 upregulation, suggesting that miR-146a-5p indirectly regulates CCNB1 through CDKN1A (Figure 4C,D and Appendix A). In parallel, miR-146a-5p downregulation in resistant cell lines led to an increase in CDKN1A expression (Figure 4E and Appendix A).

Moreover, *CDKN1A* knockdown mimicked the effect of miR-146a-5p overexpression on reducing the response to trastuzumab (*p* = 0.068 for SKBR3 and *p* = 0.030 for BT474) (Figure 4F), increasing the proliferation rate (Figure 4G), and inducing cell cycle promotion (*p* = 0.008 for SKBR3 and *p* = 0.003 for BT474) (Figure 4H). As previously shown with miR-146a-5p upregulation, *CDKN1A* knockdown also increased cell migration (*p* = 0.018) (Figure 4I,J) and angiogenesis (*p* = 0.019 for the number of branches and *p* = 0.034 for tubular length) (Figure 4K,L). The *CDKN1A* downregulation was confirmed for each experiment (Appendix A), and these results were confirmed using a second silencer (siCDKN1A#2) (Appendix A).

### 3.6. miR-146a-5p Is Enriched in Exosomes from Trastuzumab-Resistant Cells

In order to study the possibility of miR-146a-5p horizontal transference between cells, and as a consequence, the dissemination of trastuzumab resistance and miRNA expression in exosomes was examined. Exosomes from SKBR3 and SKBR3r cells were extracted following the ultracentrifugation method. The presence of exosomes was confirmed via specific protein markers (Figure 5A and Appendix A) and via visualization with transmission electron microscopy (TEM) (Figure 5B). In order to evaluate the presence of miR-146a-5p in exosomes, RNA was purified from 2 µL of extract. miR-146a-5p was demonstrated to be present in exosome samples from sensitive and resistant cell lines; however, its levels were significantly higher in exosomes from the SKBR3r cell line than from SKBR3 (*p* = 0.0001). miR-146a-5p showed a higher level of expression in exosomes compared to the original cell lines, being manifested as miR-146a-5p enrichment in exosomes (Figure 5C). Additionally, the expression of other miRNAs (miR-23b-3p and miR-26a-5p) as a control was also measured in exosomes without being detected (Appendix A).

### 3.7. Effects of Exosomes from Resistant Cells in Response to Trastuzumab by Parental Cells

We next aimed to check if the co-culture of exosomes from SKBR3r with SKBR3 could transmit the resistance to trastuzumab. First, we confirmed equal exosome uptake among the assayed conditions. For that purpose, exosomes were labeled with fluorescent DiD dye, and exosome internalization was verified using flow cytometry. The number of cells internalizing exosomes (percentage of fluorescence detected) was similar regardless of trastuzumab treatment (Figure 5D), as well as the amount of exosome uptake (mean of fluorescence) (Figure 5E).

The proliferation rate of the SKBR3 cell line in the presence of trastuzumab was evaluated when co-culturing with exosomes from SKBR3r or CM from SKBR3r. Both conditions induced a significant decrease in the effect of trastuzumab in SKBR3 (CM *p* = 0.046; exosomes *p* = 0.020) (Figure 5F). In order to confirm that the effect of exosomes from SKBR3r in trastuzumab resistance was due to miR-146a-5p transference, the combination of exosomes and the miR-146a-5p inhibitor was assayed. The inhibition of miR-146a-5p significantly reverted the resistance to trastuzumab induced by exosomes from SKBR3r in SKBR3 (*p* = 0.043) (Figure 5G).

### 3.8. Effect of Exosomes from Resistant Cells in the Migration Capacities of Parental Cells

To evaluate if exosomes could also affect the migration capacity of the cells, we performed a wound-healing assay. Exosomes from SKBR3r were added to SKBR3 cells just after realizing the wound, and the cellular migration was evaluated after 48 h. Exosome treatment increased the migration capacity of SKBR3, as shown by a faster gap-closing compared to the non-treated cells (*p* = 0.0009), reaching a similar migration ability to SKBR3r (*p* = 0.002) (Figure 6A,B). In addition, vimentin and fibronectin mRNA expression showed a significant increase in SKBR3 after co-culturing with exosomes from the resistant cells (*p* = 0.018 and *p* = 0.035, respectively), while *CDKN1A* decreased significantly (*p* = 0.0031) (Figure 6C). These results together suggest that exosomes from resistant cells are able to transmit trastuzumab resistance to parental cells through miR-146a-5p overexpression and *CDKN1A* downregulation.

### 3.9. Effect of Exosomes in Angiogenesis

To evaluate the effects of exosomes from SKBR3 and SKBR3r in cellular angiogenesis, we performed a tube formation assay in HUVEC-2 cells co-cultured with these exosomes. Exosomes from both cell lines significantly increased HUVEC-2 angiogenesis (number of branches: *p* = 0.012 for SKBR3 exosomes and *p* = 0.035 for SKBR3r exosomes; tubular length: *p* = 0.028 for SKBR3 exosomes and *p* = 0.040 for SKBR3r exosomes). As expected, this increase was greater with exosomes from the SKBR3r cells than with exosomes from SKBR3, although this difference did not reach statistical significance. (Figure 6D,E).

## 4. Discussion

HER2+ breast tumors have become an entity with a favorable prognosis and high 5-year DFS rates thanks to the incorporation of anti-HER2-targeted therapies into clinical practice. However, there are still around 20% of HER2+ BC that do not respond to trastuzumab treatment due to innate resistance or the acquisition of new molecular alterations that limit the action of the drug and confer resistance [5,6]. The main objective of this study was to evaluate the expression profiles of miRNAs involved in trastuzumab resistance. miRNAs dysregulation has been described to be common in many types of human cancer [42,43,44], and has also been related to drug resistance [45,46,47]. Our results show an important correlation between high miR-146a-5p expression, and a decreased trastuzumab response in HER2+ cell lines and lower DFS in HER2+ BC patients. miR-146a has been described as an NF-kB-dependent gene [48], and these transcription factors have been found to be constitutively activated in models of trastuzumab resistance [49]. Moreover, we have demonstrated that the overexpression of miR-146a-5p in HER2+ BC cells was able to reduce their responses to trastuzumab. Furthermore, when the acquired trastuzumab-resistant cell lines were transfected with the miR-146a-5p inhibitor, the drug response was significantly increased. Those results are consistent with the described role of miR-146a-5p in carboplatin-resistant luminal BC cells [50], as well as its implication in chemotherapy resistance, as displayed by cancer stem cells [51]. There are some conflicting data in the literature that link high levels of miR-146a-5p with reduced migration and invasion. However, these studies have been performed in triple negative or luminal BC cell lines, in contrast to our HER2+ BC model [52,53]. It is important to highlight that the analysis of the available data from METABRIC using the Kaplan-Meier Plotter for luminal and triple negative BC patients showed a protective effect of miR-146a-5p, opposite to that observed in HER2+ BC. This supports the hypothesis of a different behavior of miR-146a-5p, depending on the characteristics of the tumor cells. Furthermore, to our knowledge, this is the first study describing the role of miR-146a-5p in resistance to anti-HER2 therapy.

We found that miR-146a-5p dysregulation was impacting on the cell cycle through different gene targets. This miRNA had anteriorly been described to alter the cell cycle by affecting *CCND1* (cyclin D1) [54], *CDKN2A* [55], or the induction of the exit from the quiescent state of the mammary cancer stem cells [51]. Our results showed a significant increase in the percentage of cells in the S and G2/M phases, as well as a decrease in the G1 phase when miR-146a-5p was overexpressed. Consequently, we proposed that miR-146a-5p promotes cell cycle progression at the G1-to-S-phase transition in SKBR3r and BT474r cells as a mechanism to increase cell proliferation and resistance. This justification is coherent with our hypothesis of miR-146a-5p inducing resistance and malignancy, and it is also supported by a recent study that observes the relation between the overexpression of miR-146a-5p, increasing levels of cyclin D1, and the number of S-phase cells in cervical cancer [56].

Therefore, we analyzed *CDKN1A* and *CCNB1*, both described as putative miR-146a-5p targets. Indeed, *CDKN1A* has been proven to be a direct target of miR-146a via a dual-luciferase reporter gene assay [57]. *CDKN1A* is the gene encoding for p21, a CDK inhibitor of every cyclin/CDK complex, especially CDK2 [58]. Our findings indicate that miR-146a-5p targets *CDKN1A*, since we observe a significant inhibition following miRNA overexpression, while the oncogene *CCNB1* showed a strong increase in its expression, possibly induced by the reduction in its regulator inhibitor, p21. We have also proven this inverse and direct correlation between miR-146a-5p expression, and *CDKN1A* or *CCNB1* levels in a TCGA dataset. p21 is the protein that regulates G1 arrest and G2 arrest among other cell cycle checkpoints, while CCNB1 is a G2/M checkpoint. Lower p21 levels in the resistant cell lines as a consequence of the knockdown of the gene by miR-146a-5p could be avoiding cell cycle arrest in G1 and G2, thus increasing the S and M phases. However, we cannot distinguish between the G2 and M phases via flow cytometry due to CCNB1 modulation, causing discrepancies between the BT474 and SKBR3 cell lines, and also due to different proliferation rates between them. This decrease in p21 strongly increases cyclin B1 levels, which promotes cell cycle progression and the malignancy of resistant cell lines, avoiding apoptosis and generating genomic instability. Furthermore, miR-146a-5p gain-of-function increased migration and induced the expression of mesenchymal markers, as it is probably inducing malignancy through different pathways. We proved that *CDKN1A* loss-of-function mimicked the effect of miR-146a-5p overexpression upon reducing the response to trastuzumab, increasing proliferation and migration, and inducing cell cycle promotion on HER2+ BC cells and increasing angiogenesis in HUVEC-2 cells. These results were in agreement with the observed effects of *CDKN1A* silencing in hepatocellular carcinoma [59].

The data above support the implication of miR-146a-5p in trastuzumab resistance in HER2+ BC cells. However, there are no previous evidence of miR-146a-5p horizontal transmission with the consequent transference of trastuzumab resistance. Exosomes and other extracellular vesicles have been proven to be a potential mechanism for the horizontal cell-to-cell transmission of drug resistance [60,61,62,63]. One of the most common RNA types observed to be involved in the transmission of resistance through exosomes are miRNAs. It has been described to increase resistance to different chemotherapies in BC and other cancers via this mechanism [33,64,65,66,67,68,69]. Previous studies have demonstrated that resistance to trastuzumab could be disseminated through exosome communication involving RNAs, specifically lncRNA SNHG14 and lncRNA AFAP1-AS1 [70,71]. Similarly, miR-567 has been described to be packaged into exosomes and to revert trastuzumab resistance via the regulation of autophagy [72]. miR-146a-5p has formerly been identified in exosomes, presenting an important role in different diseases, including sepsis [73], myasthenia gravis [74], infection by EV71 [75], systemic lupus erythematosus [76], acute myocardial infarction [77], hypertension [78], or Alzheimer’s disease [79], as well as in several cancers such as non-small cell lung cancer [80], colorectal cancer [81], B-cell lymphoma [82], and adenocarcinoma of the esophagus [83]. Moreover, recent data showing that miR-146a-5p-enriched exosomes are able to reduce trastuzumab-induced oxidative stress in cardiomyocytes [84] point to the possibility that miR-146a-5p could also transfer its resistance effect using the same mechanism. The results of the present study showed that SKBR3r CM and SKBR3r exosomes partially induce trastuzumab resistance in SKBR3 cells, confirming the hypothesis of a horizontal cell-to-cell transmission of the resistance. Furthermore, the combination of SKBR3r exosomes with the inhibition of miR-146a-5p in BC cells reduced the trastuzumab resistance induced by the exosomes. Accordingly, we propose that the resistance transmitted by exosomes can be partially mediated by miR-146a-5p. Furthermore, the exosomes from resistant cells induced a significant increase in EMT markers expression, and in the migration capacity of trastuzumab-sensitive cells.

A recent study by Shan-Shan Yang demonstrates that miR-146a-5p can also promote invasion and metastasis by activating cancer associated fibroblasts (CAFs) in the tumor microenvironment via exosomes generated from BC cells [85]. Moreover, this miRNA has also been described to promote angiogenesis in HUVECs through different mechanisms [86,87]. miR-146a-5p inhibition impairs tube formation and migration in endothelial cells [88], and its neutralization reduces angiogenesis in vitro and in vivo in a cancer colon mouse model [89]. Since exosomes have been proven to promote angiogenesis in cancer [90], typically via miRNAs [91,92,93], we hypothesized that exosomes derived from our BC cell lines, which contain miR-146a-5p, would also be inducing angiogenesis as one of its roles. Our findings demonstrate that the angiogenic capacity of HUVEC-2 was significantly enhanced by CM enriched with miR-146a-5p in terms of branches and tube length. Furthermore, the co-culture of HUVEC-2 with exosomes reproduced this effect, and as expected, angiogenesis was higher with SKBR3r exosomes than with SKBR3 exosomes, concurring with their higher levels of miR-146a-5p. All of this is consistent with the roles of exosomes in cancer progression and aggressiveness, as previously described in the literature [94,95,96,97,98,99].

The results of this study demonstrate for the first time the implication of miR-146a-5p in trastuzumab resistance in HER2+ BC models. The gain and loss of miR-146a-5p modulates the resistance to trastuzumab through cell cycle regulators. It should be noted that patients with shorter DFS and OS present higher levels of miR-146a-5p. Our results also confirmed that exosomes have an important role in the transmission of trastuzumab resistance, partially performed by miR-146a-5p, and also suggest an effect of tumor progression via the modulation of the microenvironment.

The determination of miR-146a-5p expression in primary tumors from HER2+ BC patients could be useful for the early identification of trastuzumab resistance. Moreover, exosomes have been shown to be a valuable clinical tool, both as diagnostic biomarkers and in monitoring a treatment response [34,35]. Furthermore, the repression of miR-146a-5p by antagomir or similar technologies may be a therapeutic option for increasing the response to trastuzumab and in inhibiting the spread of drug resistance.

## 5. Conclusions

The overexpression of miR-146a-5p in HER2+ BC associates with trastuzumab resistance. The gain- and loss-of-function of this miRNA modulates trastuzumab resistance in vitro. Furthermore, elevated miR-146a-5p levels in the primary tumor tissues of HER2+ BC patients are associated with a worse prognosis. The accumulation of a high proportion of the cell population in the S and G2/M phases of the cell cycle, and a decrease in the expression of CDKN1A are the potential mechanisms of action, together with the increase in migration and angiogenesis. Moreover, the exosomes of SKBR3r cells contain high miR-146a-5p levels and are able to reduce the effect of trastuzumab on parental cells, increasing their expression of EMT markers and cell migration. Moreover, the exosomes of SKBR3r cells increase the angiogenic ability of HUVEC-2 cells as aggressive properties. Altogether, our data suggest miR-146a-5p levels are a potential biomarker of trastuzumab response, and a valuable target for future anti-HER2 treatment strategies.

## Figures and Tables

**Figure 1 cancers-15-02138-f001:**
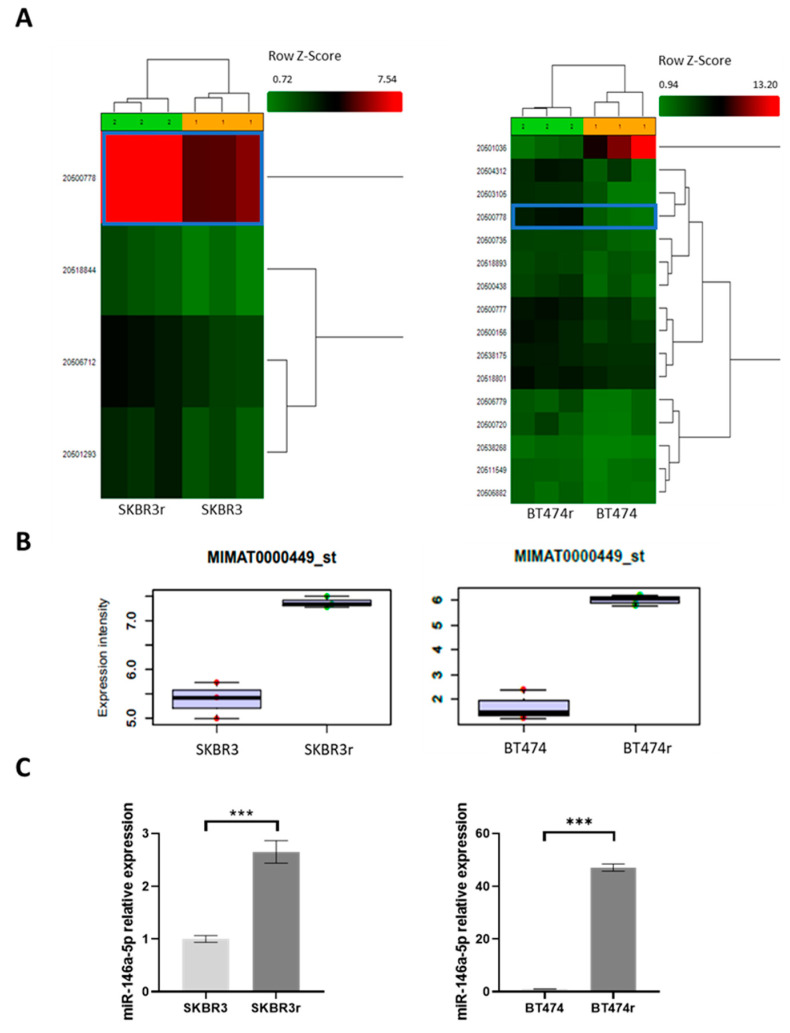
miRNA differential expression between parental and trastuzumab-resistant cell lines. (**A**) The heat map shows significant differences in miRNA expression between SKBR3/SKBR3r and BT474/BT474r analyzed via the Affimetrix human miRNA microarray GeneChip miRNA 4.0 Array, and their hierarchical clustering. miR-146a-5p expression is highlighted with a box. Three repeated samples were used for each cell type. (**B**) Expression intensity of miR-146a-5p (MIMAT0000449_st) in SKBR3/SKBR3r and BT474/BT474r samples of the GeneChip miRNA 4.0 Array of the Affymetrix microarray. (**C**) Expression levels of miR-146a-5p in cells were measured via RT-qPCR for validation. GAPDH was used as an endogenous control. RT-qPCR was performed in technical and biological triplicates. *** *p*-value < 0.001.

**Figure 2 cancers-15-02138-f002:**
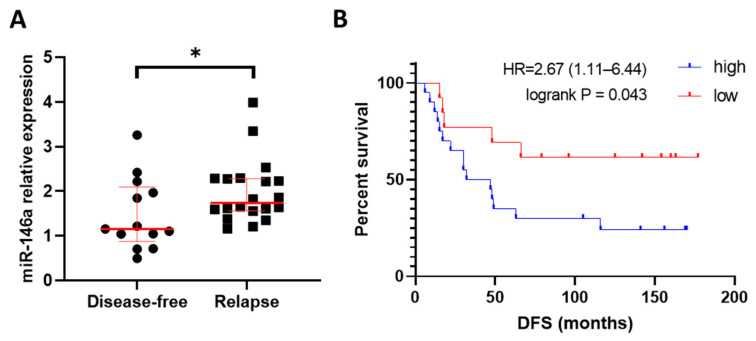
Analysis of the expression of miRNA-146a-5p in HER2+ breast cancer patients. (**A**) Levels of miR-146a-5p expression measured in disease-free HER2+ BC patients (n = 13) and relapsed HER2+ BC patients (n = 20) via RT-qPCR. miR-16a was used as endogenous miRNA. (**B**) Patients (n = 33) were classified based on median miR-146a-5p expression in tumor samples, and DFS Kaplan-Meier curve was plotted (*p*-value = 0.043). * *p*-value < 0.05. HR: Hazard ratio; DFS: Disease-free survival.

**Figure 3 cancers-15-02138-f003:**
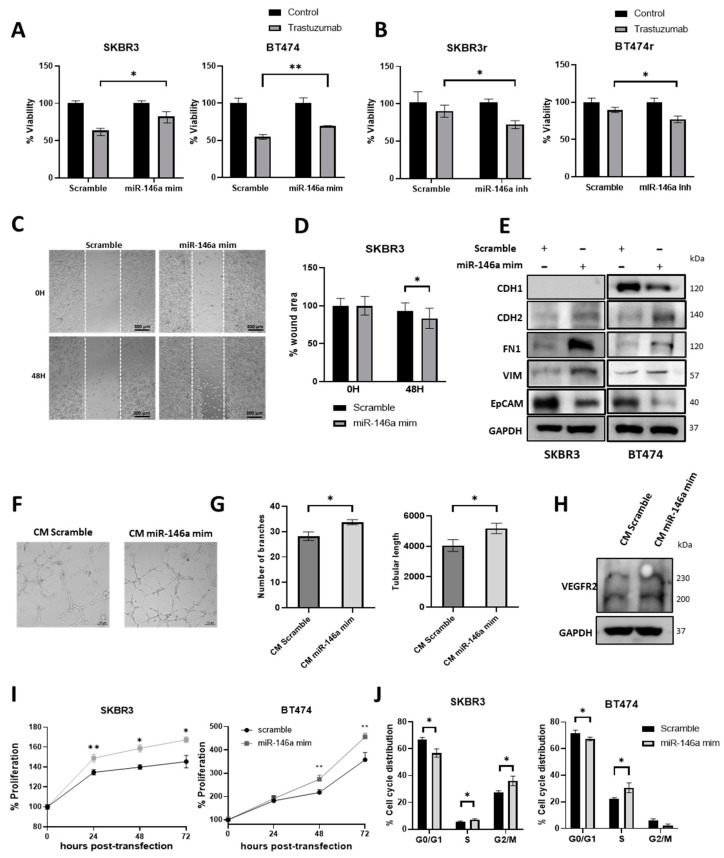
Analysis of miR-146a-5p effect in response to trastuzumab, migration, angiogenesis, proliferation, and cell cycle. (**A**) Viability analysis of SKBR3 and BT474 transfected with 50nM miR-146a-5p mimic (**B**) SKBR3r and BT474r transfected with 50 mM miR-146a-5p inhibitor (**B**) with or without treatment with 15 µg/mL of trastuzumab for 7 days. (**C**) Representative images of wound-healing cell migration assay of SKBR3 transfected with 50 nM miR-146a-5p mimic. Magnification: 10×. Scale bar: 300 µm. (**D**) Quantification of the remaining wound area at 0 and 48 h under conditions indicated above. (**E**) Expression of epithelial-to-mesenchymal transition markers (CDH1, CDH2, FN1, and VIM) and EpCAM in SKBR3 and BT474 via Western blot after 50 nM miR-146a-5p mimic transfection. GAPDH was used as loading control. (**F**) Representative images of in vitro angiogenesis assay of HUVEC-2 cells (10× magnification). (**G**) Representations of the number of branching points and the relative tubular length 8 h after addition of CM of SKBR3 transfected with miR-146a-5p or scramble control. (**H**) Expression of VEGFR2 in HUVEC-2 after 24 h of treatment with CM of SKBR3 transfected with miR-146a-5p mimic or scramble control. (**I**) Proliferation analysis of SKBR3 and BT474 transfected with 50 nM miR-146a-5p mimic or scramble at 0, 24, 48, and 72 h. (**J**) Cell cycle distribution of SKBR3 and BT474 cells overexpressing miR-146a-5p. Propidium iodide was used for staining, and the analysis was performed via flow cytometry. Each experiment was performed in technical and biological triplicate. * *p*-value < 0.05; ** *p*-value < 0.01. CM: Conditioned medium; CDH1: E-Cadherin; CDH2: N-cadherin; FN1; Fibronectin; VIM: Vimentin; EpCAM: Epithelial cell adhesion molecule. The uncropped blots are shown in Appendix A.

**Figure 4 cancers-15-02138-f004:**
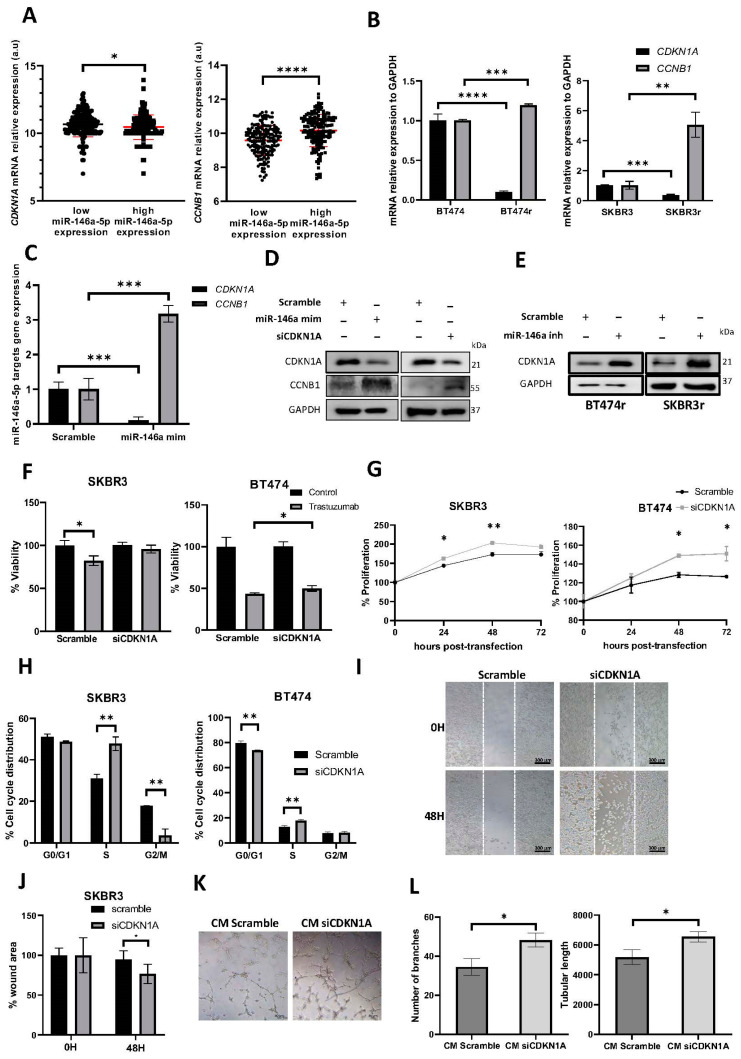
CDKN1A modulates the response to trastuzumab, proliferation, cell cycle, migration, and angiogenesis. (**A**) Data from TCGA dataset (n = 333) were classified based on median miR-146a-5p expression in breast tumor sample, and *CKDN1A* and *CCNB1* expression were represented. (**B**) An analysis via RT-qPCR of *CDKN1A* and *CCNB1* expression in BT474/BT474r, and SKBR3/SKBR3r cell lines, (**C**) or in BT474 transfected with miR-146a-5p mimic. GAPDH was used as endogenous mRNA. (**D**) Analysis via Western blot of CDKN1A and CCNB1 in BT474 after miR-146a-5p mimic or small-interfering (siRNA) targeting *CDKN1A* (siCDKN1A) transfection, (**E**) and in BT474r and SKBR3r after miR-146a-5p inhibitor transfection. GAPDH was used as a loading control. (**F**) Viability analysis of SKBR3 and BT474 transfected with siCDKN1A, treated or not with 15 µg/mL of trastuzumab for 7 days. (**G**) Proliferation analysis of SKBR3 and BT474 cell lines after 0, 24, 48, and 72 h after siCDKN1A transfection. (**H**) Cell cycle distribution of SKBR3 and BT474 cells transfected with siCDKN1A. (**I**) Wound-healing cell migration assay of SKBR3 transfected with siCDKN1A, 5× Magnification. Scale bar: 300 µm. (**J**) Quantification of the wound remaining area at 0 and 48 h post-transfection. (**K**) Representative images of in vitro angiogenesis assay of HUVEC-2 cells 8 h after addition of CM after 24 h of SKBR3 transfected with siCDKN1A or scramble control. Magnification: 10×. Scale bar: 50 µm. (**L**) Representation of the number of branching points and the relative tubular length. Each experiment was performed in technical and biological triplicate. * *p*-value < 0.05; ** *p*-value < 0.01; *** *p*-value < 0.001, **** *p*-value < 0.0001. CM: Conditioned medium. The uncropped blots are shown in Appendix A.

**Figure 5 cancers-15-02138-f005:**
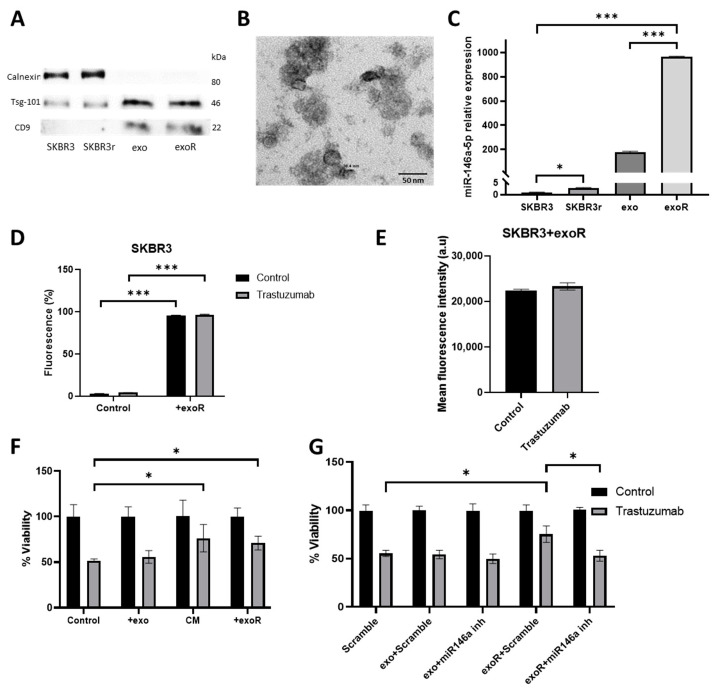
Characterization of SKBR3 exosomes and their effect in trastuzumab response. (**A**) Western blot was performed for measuring the presence of TSG-101 and CD9 typical from exosomes, and the absence of calnexin as a cellular control. (**B**) Representative image of exosomes observed via TEM (microscope JEOL1010, 60,000× Magnification) (scale bar: 50 nm). (**C**) Analysis via RT-qPCR of miR-146a-5p expression in exosomes from SKBR3 (exo), exosomes from SKBR3r (exoR) line, and from SKBR3 and SKBR3r cell lines. miR-16a was used as endogenous miRNA. (**D**) Percentage of fluorescent cells detected via flow cytometry after exosome internalization assay with DMEM F-12 alone or exoR containing (**E**), and mean of fluorescence detected in cells treated or not with trastuzumab. (**F**) Viability assay of the SKBR3 cell line co-cultured with 50% of CM from SKBR3r or exoR, treated with or without 15 µg/mL trastuzumab for 7 days. (**G**) Viability assay of the SKBR3 cell line co-cultured with exoR after transfection with miR-146a-5p inhibitor with or without 15 µg/mL trastuzumab for 7 days. Each experiment was performed in technical and biological triplicate. * *p*-value < 0.05; *** *p*-value < 0.001. The uncropped blots are shown in Appendix A.

**Figure 6 cancers-15-02138-f006:**
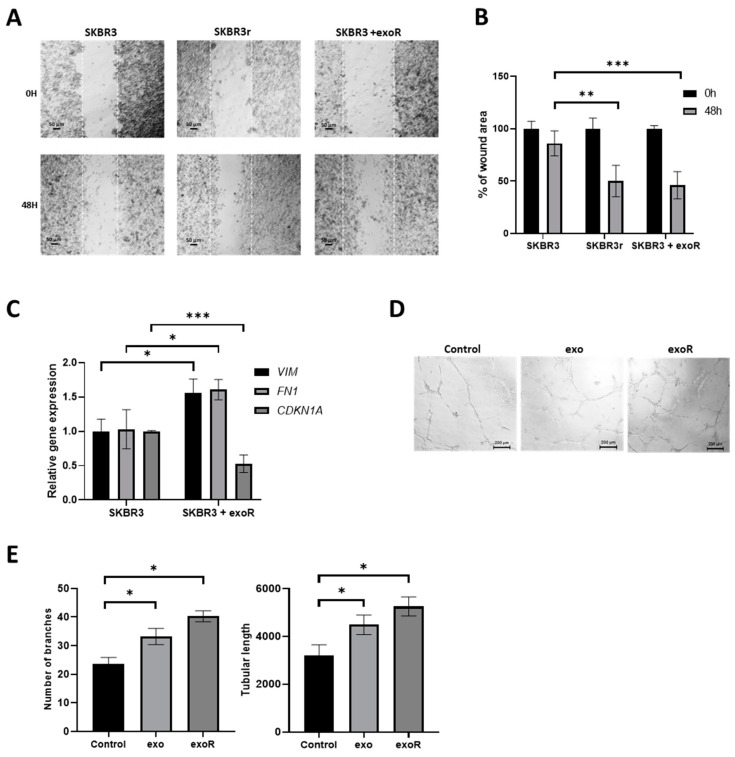
Effects of exosomes in cell migration and angiogenesis. (**A**) Representative images of wound-healing cell migration assay of SKBR3, SKBR3r, and SKBR3 co-cultured with exosomes from SKBR3r (exoR). Magnification: 10×. (**B**) Quantification of the wound remaining area at 0 and 48 h. (**C**) Expression of EMT markers (VIM and FN1) and miR-146a-5p target CDKN1A, measured via RT-qPCR after co-culture of SKBR3 with exoR. GAPDH was used as endogenous mRNA gene, and the expression was calculated using the ΔΔCt method. (**D**) Representative images of in vitro angiogenesis assay of HUVEC-2 cells 8 h after addition of exo or exoR. Magnification: 5× (scale bar: 200 µm). (**E**) Representations of the number of branching points and the relative tubular lengths of HUVEC-2 cells 8 h after exo or exoR addition. Each experiment was performed in technical and biological triplicate. * *p*-value < 0.05; ** *p*-value < 0.01; *** *p*-value < 0.001. VIM: Vimentin; FN1: Fibronectin.

**Table 1 cancers-15-02138-t001:** Clinicopathological characteristics of HER2+ BC patients.

Characteristics	HER2+ BC Patients
Number	33
Median age in years (range)	51.5 (35–70)

HR	
HR+	17 (51.52%)
HR−	16 (48.48%)
Stage, N (%)	
I	3 (9.09%)
II	23 (69.70%)
III	6 (18.18%)
IV	1 (3.03%)
Grade	
II	11 (33.33%)
III	21 (63.64%)
NA	1 (3.03%)
Treatment	
Adjuvant	18 (54.55%)
Neoadjuvant	15 (45.45%)
Response	
Disease-free	13 (39.40%)
Relapsed	20 (60.60%)
Median follow up (months)	97
Median DFS	49 (6–177)
Median OS	96 (7–177)

NA: Not available.

## Data Availability

The microRNA data have been uploaded to the page: https://submit.ncbi.nlm.nih.gov/geo/ (GSE197822) (accessed on 25 December 2022), with the names mi1_(miRNA-4_0) CEL to mi12_(miRNA-4_0). CEL for the data of all 12 samples and PCPE_affy_miRNA_.xls.

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
