# Peer review of "miR-146a-5p Promotes Angiogenesis and Confers Trastuzumab Resistance in HER2+ Breast Cancer"

_cancers, 2023, doi:10.3390/cancers15072138_

Round 1
Reviewer 1 Report (Previous Reviewer 2)
The authors addressed several major concerns, significantly improving the quality of the manuscript. However, some important points remained unaddressed:
1) MTT assay should be paired with another viability assay, like crystal violet or colony assay to distinguish proliferation inhibition from cell death. Also, BrdU or Ki67 should be used for proliferation.
2) About the treatments with miR-146a-5p mimetic/inhibitor I previously asked:
The authors performed a transient transfection with miR-146a-5p mimetic/inhibitor and then proceed to analyze cell viability after 7 days of treatment with trastuzumab. I wonder how much miR-146a-5p mimetic/inhibitor is still present after 7 days. The authors should perform these experiments with shorter treatments time points and show miR-146a-5p expression in each condition.
The authors replied: "The aim of this experiment was to check if the increase o reduction of miR-146a-5p levels modified the resistance of cells. We evaluated the viability at 7 days because it is the time that has been set up for us and others authors [1] to assess the resistance, by which cell lines with a fold decrease in growth rate of ≥1.2 or a decrease in colony number of ≥20% after 7 days in response to trastuzumab were considered sensitive. We have also published several studies with these proceedings [2, 3].For our experiments, we checked the miR-146a-5p levels at 48 hours after transfection to know if the experiment was working properly. If this happens, we expect to see changes at 7 days regardless of how long the miRNA level change is maintained. The transfected miRNA will bind mRNAs of its gene targets and this will eventually affect to differential translation, and so differences in protein level. This physiological effect that may affect resistance is what we aim to study at 7 days, and doesn’t depend on the remaining miRNA levels at that moment anymore. In fact, the change that we observe in trastuzumab effect at 7 days in this experiment must be caused by the downstream pathway of our miRNA, as we are treating clones only differing by this factor".
I get the point of the 7day experimental endpoint necessary for resistance. However, how can we be sure that an agent causes sensitivity or resistance without being there? It's like assuming that a car accident was caused by inclement weather weeks after the ice melted. Even in the case of a translational mechanism, changes are visible at 48-72h. A simple way to address this question is to manipulate miR-146a-5p expression once cells are already resistant and prolong the treatment for additional 48-72 hrs post miR-146a-5p up/down. Same is valid for migration.
3) Authors said "We didn’t detect E-cadherin in SKBR3 cell line as this cell line presents loss of CDH1 as a mechanism for tumour progression"
Hence, the authors need to show an alternative epithelial marker (e.g.EpCAM) to prove EMT.
4) authors said "We have followed the reviewer’s indications showing CDKN1A and FN1 expression (Fig. 6 C). CTNNB1 levels did not change significantly may be due to the low levels of CTNNB1 detected, as SKBR3 has been described to be CTNNB1-negative [7]."
Again, either SKBR3 cells aren't the proper model or CTNNB1 is not a generalized mechanism. The authors should tone down the relevance of CTNNB1 as downstream effector.
Author Response
1.Regarding the reviewer’s recommendation, we have also performed a 7 days viability assay by flow cytometry using propidium iodide. See figure in the attached document.
We can observe trastuzumab is causing cell death in sensitive cell line, that can be partially reverted with miR-146a-5p overexpression. On the other hand, resistant cell line shows no cell death when treated with trastuzumab unless miR-146a-5p is inhibited, partially reverting the resistance of the cells.
Trastuzumab also induces cycle arrest by increases G1 phase and reduces S and G2/M phase (Díaz-Gil. L, et al.) so the effect we detect by MTT is a combination of both cell death and proliferation inhibition by trastuzumab.
We also addressed Ki67 expression for proliferation as proposed by the reviewer. We demonstrated that trastuzumab is not only promoting cell death but also inducing proliferation arrest. miR-146a-5p inhibitor also inhibited cell proliferation and had a synergic effect with trastuzumab treatment. See figure in the attached document
- We understand the reviewer’s concerns, so we have checked miRNA-146a-5p expression also after 7 days of transfection (Supp. Fig. 4B). Thus, we demonstrate that the transfected miRNA is still overexpressed after 7 days and is responsible for the changes in trastuzumab effect measured at day 7.
- Following the reviewer’s advice, we have included EpCAM expression in Figure 3E and Supp. Figure 5A. Overexpression of miR-146a-5p reduced EpCAM expression in parental cells compared to transfection with scramble.
- We appreciate the reviewer’s suggestion and have suppressed CTNNB1 from the manuscript, both from figures and text references.
Bibliography:
Díaz-Gil. L, et al. Modelling hypersensitivity to trastuzumab defines biomarkers of response in HER2 positive breast cancer. J Exp Clin Cancer Res. 2021 Oct 7;40(1):313.

Reviewer 2 Report (Previous Reviewer 3)
The authors largely answered my questions except for Questions 5 and 6. In Supp Fig 7D, again, there was no cell proliferation for SKBR3-scramble cells, even slightly decrease comparing with 0 hr, which suggested non-specific toxicity of scramble siRNA under this condition. These results were not consistent with Fig 3I, SKBR-Scramble cells. The authors need to either repeat the experiments with optimized conditions or discuss the inconsistency in discussion section.
Author Response
We appreciate the reviewer’s recommendation and have repeated the experiment for figure 7D, showing a better proliferation curve.
This manuscript is a resubmission of an earlier submission. The following is a list of the peer review reports and author responses from that submission.
Round 1
Reviewer 1 Report
The author reported their findings that miR-146a-5p was transferred through exosomes of breast cancer cells and plays an important role in HER2+ breast cancer resistant to trastuzumab treatment. However, the manuscript need to be studied more carefully and make some corrections,
1) In miRNA array assay, the author identified that miR-146a-5p was overexpressed in drug resistant breast cancer cells. However, when look into the expression of miR-146a-5p between two cell lines, the expression of miR-146a-5p in parental SKBR3 is almost same as the drug resistant BR474; and the fold change between SKBR3 and SKBR3R is less than two times while fold change between BR474 and BR474R is over 50 times in qRCR results, which suggests that BR474 might be a better cell for further experiments. Instead, the author used SKBR3 cells in most of the experiments.
2) For cell proliferation results, does none of miR-146a mimic, miR-146a inhibitor, siCDKN1A affect the growth of cancer cells? All the bar chart of control treatment % of cell growth is around 100%.
2) The results and figure legend are missing the whole part of figure 3G-J, figure 4F-I
3) Figure 4A, no obvious difference in CDKN1A were observed in the figure. How the statistics analysis was done?
4) Result 3.5. In this part, the author transfect SKBR3 cells to overexpress and inhibit miR-146a-5p and compare the CDKN1A expression. Since, the SKBR3r was identified to have miR-146a-5p overexpression. Why not just compare the CDKN1A in SKBR3 and SKBR3r? And would it be better to transfect SKBR3r or miR-146a-5p transfected SKBR3 with siCDKN1A instead of simply transfect parental SKBR3 cells to study if the cell proliferation modulated by miR-146a is through regulating CDKN1A.
Reviewer 2 Report
The article “miR-146a-5p confers resistance to trastuzumab in HER2+ breast 2 cancer” by Cabello and Torres-Ruiz et al. describes a novel role of the exosomal miR-146a-5p in inducing Trastuzumab resistance in HER2+ breast cancer. With gain and loss of function experiments the authors aim at demonstrating that miR-146a-5p present in exosomes regulates trastuzumab resistance via modulating cell proliferation, cell cycle and angiogenesis.
However, several minor and major points jeopardize the impact of this study and need to be addressed prior to be considered for publication in Cancers again:
Line 77 use “among” instead of “between”
Line 173-174 specify concentrations of uranyl acetate and lead citrate
Line 179 “at 40%...amplitude?”
Western blot methods: please add quantity of protein used and antibody dilutions used.
Wound healing assay: After 48 hours cells have proliferated and this might jeopardize the results of migration. Especially in this case where “miR-146a-5p overexpression in SKBR3 and BT474 cell lines increases the proliferation rate of the cells in time-dependent manner (Fig. 3C). Moreover, we analysed miR-146a-5p effect on migration and EMT”. The authors need to inhibit proliferation with mitomycin or actinomycin C, before creating the gap. Same valid for siCDKN1A wound healing (figure 4F).
Line 278 I would specify “parental and resistant” instead of “partner”
MTT assay should be paired with another viability assay, like crystal violet or colony assay to distinguish proliferation inhibition from cell death. Also, BrdU or Ki67 should be used for proliferation. As it is, Y axis of the MTT assay graphs should say “viability”.
Treatments with miR-146a-5p mimetic/inhibitor (Figure 3):
The authors performed a transient transfection with miR-146a-5p mimetic/inhibitor and then proceed to analyze cell viability after 7 days of treatment with trastuzumab. I wonder how much miR-146a-5p mimetic/inhibitor is still present after 7 days. The authors should perform these experiments with shorter treatments time points and show miR-146a-5p expression in each condition. This point is confusing because in the method it is explained that cell lines are transfected and then treated with trastuzumab for 7 days, but then in the legend of figure 3 they say “analysis of SKBR3 and BT474 cell lines after 0, 24, 48 and 72 hours”. However, only one graph per condition is present and it’s not clear which endpoint is.
The authors also stated “miR-146a-5p overexpression in SKBR3 and BT474 cell lines increases the proliferation rate of the cells in time-dependent manner (Fig. 3C)” but panel 3C is referring to wound healing. Perhaps a panel is missing. Check figure, legend and result description.
Figure 3E. E-cadherin is not detected in SKBR3. Therefore authors should use a higher amount of protein or choose another epithelial marker, such as EPCAM. The authors should indicate which time point post transfection was used. Also, how many times was this WB performed? Is there a quantification? At least two independent WBs or independent samples (biological reps) should be assayed.
CM experiment Figure 3F, G, H
These data are not described at all. Again, numbering is all scrambled in the text. Please make sure to describe all the data.
Line 314 use “software” instead of “softwares”.
Cell Cycle
Authors refer to Cell cycle data in Figure 4A which is instead figure 3J. All figures are scrambled. Please, check all the figures and figure legends and make sure they are consistent. Also, miR-146a-5p overexpression does not cause G2/M increase in BT474. The authors should comment on that. Also, flow plots must be reported along with their quantification. Same for figure 4E. Also explain discrepancies between SKBR3 and BT474.
siCDKN1A
Since the authors claim that both CDKN1A and CCNB1 are miR-146a-5p targets, they should use more than one siRNA (at least 2) for both CDKN1A and CCNB1.
Lines 366-368 “Additionally, the expression of other miRNAs (miR-23b-3p and miR-25a-5p) as control were also measured in exosomes without being detected (data not shown)”. Authors should use other markers as control, and show the data in supplementary.
Exosome experiments
How can the authors be sure that the uptake of exo and exoR is the same? Also, how can they be sure that co-treatment with trastuzumab does not influence exosomes uptake, too? The author should show some kind of control (e.g. exosomes with fluorescent probe) to ensure data aren’t biased by differential exosome uptake, but only by their cargo.
Also in Figure 5D and 5E the authors should include a control with exo from parental cells.
Figure 5C. The authors should show CDKN1A expression and explain why FN1 and CTNNB1 did not change significantly. This is a discrepancy with Figure 4B
As for the angiogenesis experiments, how does miR-146a-5p angiogenesis influence Trastuzumab resistance? The authors should probably do a co-culture with HUVEC and SKBR3 (see protocol example https://journals.plos.org/plosone/article?id=10.1371/journal.pone.0253258) +/- miR-146a-5p to show whether it matters or not. Otherwise they should indicate angiogenesis as a factor independent of Trastuzumab resistance in the manuscript title. Example: miR-146a-5p promote angiogenesis and confers trastuzumab resistance in HER2+ breast 2 cancer.
General points that need be addressed throughout the entire manuscript:
Conceptual comments
· The entire article is based on the fact that miR-146a-5p is more abundant in Trastuzumab-resistant cells and that CDNK1B, CTNNB1 are its likely target genes. To my point of view, the main interest should be to know which mechanisms induce higher miR-146a-5p levels in resistant cells. Is it transcriptional regulation? If yes, which putative TFs are regulating miR-146a-5p transcription? Or is it a post-translational regulation? Or both? Authors should discuss this aspect.
· The main mechanism of action of Trastuzumab is ADCC (antigen-dependent cell cytotoxicity). I strongly encourage the authors to analyze the effect of exoR and miR-146a-5p mim/inh with this assay.
Scientific rigor and visualization comments:
· Check all the figures as many of them are scrambled and do not match with legends/description.
· Images are blurry due to low resolution. Please improve image quality.
· In all the figures, both error limits (top and bottom) should be displayed along with all the replicate points. Error bars type (SD, SEM), n of replicates and experiments should be specified in all the figure legends.
· All results should be consequentially described as they are displayed and vice versa.
· All blots need molecular weight indicators.
· Check that all the methods are included and sufficiently described so that others can replicate the experiments.
· Check throughout for typos, abbreviations and misspellings.
Reviewer 3 Report
Major comments:
1. The manuscript is not well prepared. With some figure panels in Figure 3 and 4, there are no corresponding figure legends and wrongly cited in main text. In addition, some labels are not clear in Figure 3. All of these make it hard to read the manuscript.
2. With exosome studies, normal FBS was used in this study, not the exosome-depleted FBS. This may confound all the results generated from exosome experiments.
3. For many experiments in this work, triplicate was performed. However, the authors did not specify if it is technical replicate or biological replicate. Biological replicates should be performed.
4. Western blotting results need to be quantified and repeats need to be performed. GAPDH was over-exposed in most of western blotting results making the results hard to interpret. Especially for Figure 4B, GAPDH levels were different and densitometry values are required to interpret the results.
5. For knockdown experiments using siRNA, only one siRNA was used and concentration was too high, 100 nM. With these conditions, it is hard to exclude off-target effect.
6. For Figure 3I (In figure panel, not found in main text), the experiment was not well performed. With scramble condition, there was no increase of BT474 cells proliferation at 24 and 48 hours comparing with 0 hour, and only mild increase of SKBR3 cells proliferation from 24 to 72 hours but a big increase from 0 to 24 hours. These results are not normally growth curves look like, suggesting scramble condition is highly toxic or cells used in these experiments are not good. Conditions need to be optimized further, and mock control or parental cell control needs to be included.
7. IC50 values are calculated based on results in Supplementary Figure 3. However, the doses of trastazumab was not well designed and it is hard to figure out accurate IC50 values from those doses. The authors can use a different way to address the sensitivity of cell lines to the treatment.